# Potential of Multiscale Astrocyte Imaging for Revealing Mechanisms Underlying Neurodevelopmental Disorders

**DOI:** 10.3390/ijms221910312

**Published:** 2021-09-24

**Authors:** Takuma Kumamoto, Tomokazu Tsurugizawa

**Affiliations:** 1Developmental Neuroscience Project, Department of Brain & Neurosciences, Tokyo Metropolitan Institute of Medical Science, Tokyo 156-8506, Japan; 2Human Informatics and Interaction Research Institute, National Institute of Advanced Industrial Science and Technology (AIST), 1-1-1 Umezono, Tsukuba 305-8568, Japan; 3Faculty of Engineering, Information and Systems, University of Tsukuba, Tenoudai 1-1-1, Tsukuba 305-8573, Japan; 4Department of Neuroscience, Jikei University School of Medicine, 3-25-8 Nishishinbashi, Tokyo 105-8461, Japan

**Keywords:** astrocyte, development, glymphatic system, neuropsychiatric disease, microscopic imaging, functional imaging

## Abstract

Astrocytes provide trophic and metabolic support to neurons and modulate circuit formation during development. In addition, astrocytes help maintain neuronal homeostasis through neurovascular coupling, blood–brain barrier maintenance, clearance of metabolites and nonfunctional proteins via the glymphatic system, extracellular potassium buffering, and regulation of synaptic activity. Thus, astrocyte dysfunction may contribute to a myriad of neurological disorders. Indeed, astrocyte dysfunction during development has been implicated in Rett disease, Alexander’s disease, epilepsy, and autism, among other disorders. Numerous disease model mice have been established to investigate these diseases, but important preclinical findings on etiology and pathophysiology have not translated into clinical interventions. A multidisciplinary approach is required to elucidate the mechanism of these diseases because astrocyte dysfunction can result in altered neuronal connectivity, morphology, and activity. Recent progress in neuroimaging techniques has enabled noninvasive investigations of brain structure and function at multiple spatiotemporal scales, and these technologies are expected to facilitate the translation of preclinical findings to clinical studies and ultimately to clinical trials. Here, we review recent progress on astrocyte contributions to neurodevelopmental and neuropsychiatric disorders revealed using novel imaging techniques, from microscopy scale to mesoscopic scale.

## 1. Introduction

The adult human brain contains roughly as many glial cells as neurons (84.6 ± 9.8 billion versus 86.1 ± 8.1 billion), and glial cells outnumber neurons by 3.76-fold in the cerebral cortex [1], underscoring their developmental and physiological importance. In mouse cortex, the ratio of astrocytes to neurons is even greater (8.4:1) across all regions of gray matter (GM) [2]. Astrocytes have multiple biological functions, including organization of the blood–brain barrier [3], clearance of metabolites [4], modulation of synaptic function to control NMDAR-dependent plasticity [5], and perisynaptic glutamate and potassium clearance [6]. Astrocyte dysfunction is implicated in numerous neurodevelopmental, neurodegenerative, and neuropsychiatric disorders such as Alexander’s disease [7], Rett syndrome [8,9,10,11], fragile X syndrome [12], epilepsy [13,14,15], and Huntington’s disease (HD) [16]. To understand the causes of these congenital diseases and create effective treatments, it is essential to fully elucidate the mechanisms of astrocyte development and the consequences of these developmental processes on brain structure and function. In the present review, we describe some of the latest technological advances in mesoscopic and microscopic imaging that are broadening our understanding of astrocyte development and the contributions of astrocyte dysfunction to neurodevelopmental and neuropsychiatric diseases.

## 2. Astrocyte Development in Cerebral Cortex

In the past three decades, technologies such as mouse transgenics, gene expression analyses, high-throughput single-cell DNA and RNA sequencing, and multiscale imaging analyses have vastly expanded our understanding of the development and functions of astrocytes. Morphological, lineage, and gene expression analyses, for example, have enabled the distinction of two distinct astrocyte subtypes, (1) protoplasmic and (2) fibrous [17], in rodents, as well as two further subclasses, (3) interlaminar astrocytes and (4) varicose-projection astrocytes, in humans [18,19]. Protoplasmic astrocytes are predominant in GM, whereas fibrous astrocytes are predominant in white matter (WM), and each type exhibits distinct functions tailored to the local environment [19,20]. Astrocytes pass through highly coordinated developmental stages, and it is expected that dysfunction during specific stages and in specific regions will result in a spectrum of developmental, behavioral, and cognitive abnormalities after birth. Therefore, it is critical to understand the processes governing astrocyte development at the genetic and molecular levels and how disruption of these processes can manifest in specific brain deficits. In this section, we describe the latest findings on when, where, and how astrocytes are produced and distributed in the developing cerebral cortex.

### 2.1. Spatial Origins and Heterogeneity of Astrocytes

In addition to the morphological subtyping described above, clustering has recently been used to classify subtypes using single-cell analysis. For example, astrocytes in adult mouse telencephalic and non-telencephalic regions can be divided into seven subtypes, with fibrous and protoplasmic astrocytes in the telencephalon reclassified as ACTE1 and ACTE2, respectively [21]. In the mammalian cerebral cortex, astrocytes are derived from radial glia cells located in the ventricular zone (VZ) and ventral forebrain during development [16,22,23,24,25]. There are two broad morphological subtypes of astrocytes, protoplasmic and fibrous, in the cerebral cortex, and each may have different functions that depend on distribution and gene expression profile [22]. Furthermore, subtype-specific dysfunction may lead to unique neurological deficits.

The developing cerebral cortex is organized into six neuronal layers distinguished by developmental order, neuronal subtype distribution, and circuit characteristics [26], and these layers are further subdivided into tangential compartments according to the thickness of the layer, each having a different functional circuit (termed “arealization”) [27]. Findings that astrocytes are nonrandomly distributed among radial layers and tangential cortical areas suggest the possibility of highly localized and specialized functions for these cells. Batiuk et al. distinguished five astrocyte subtypes (AST1-5) in cerebral cortex and hippocampus of postnatal day 56 (P56) mice by single-cell RNA sequencing of ACSA-2-PE immunolabeled cells [28], and they found that each subtype was differentially distributed within these brain regions. Similarly, Lanjakornsiripa et al. found layer-specific morphological differences among astrocyte populations between the cortical upper layer (UL) and deep layer (DL), such as distinct cell orientation, territorial volume, and arborization. Furthermore, RNA sequencing of manually dissected UL and DL indicated molecular and morphological differences between layers [29]. In *Reeler* and *Dab1* KO mice, mutants in which the six-layer laminated structure of the cortex is abnormal, the morphological and molecular differences between UL and DL astrocytes were absent, suggesting that astrocyte phenotype distribution depends on the establishment of neuronal layer identity during development [29]. Bayraktar et al. established a large-area spatial transcriptomic map (LaST) displaying astrocyte layers in three regions of mouse cerebral cortex by immunohistochemical staining and single-molecule fluorescence in situ hybridization (smFISH) of 46 candidate astrocytic genes with further confirmation by single-cell RNA sequencing and spatial reconstruction analysis [30]. Importantly, this LaST map revealed multiple astrocyte subtypes with layer-specific distributions, as well as tangential differences among areas at the molecular level. Furthermore, the authors found that the identities of these astrocyte layers were established by postmitotic neuronal cues as suggested by the altered distributions in *Reeler* and *Satb2* cKO mice. Analysis of *Satb2* cKO mice also revealed that acquisition of superficial layer astrocyte identity required layer 4 neuronal identity, and that the spatial distribution of layer-specific astrocytes was inverted in *Reeler* mice concomitant with the change in neuron distribution [30].

### 2.2. Temporal Fate Specification of Astrocytes

In mice, cortical neurogenesis begins with the production Cajal–Retzius cells in the marginal zone and the subsequent development of first-born DL neurons around E11, followed sequentially by production of UL neurons until E16 and then gliogenesis [2,22,31]. Neurons first migrate along the processes of radial glial cells (RGCs) extending across the cortex, and both later-born neurons and astrocytes appear to be derived from RGCs. For instance, single RGCs isolated from the cortical VZ by fluorescence activated cell sorting and cultured to monitor clonal maturation differentiated into both neurons and astrocytes [32]. A lineage study of *Thy1.2-Cre* mice also suggested that astrocytes were generated from RGCs after neurogenesis [2]. Subsequent studies examined the timing and factors controlling the transition from neurogenesis to gliogenesis. Shen et al. examined the onset of gliogenesis from RGCs using the Mosaic analysis with double markers (MADM) system [33] and found that differentiation of cortical astrocytes from RGCs occurs between E16 and E17 in well-defined proportions, with 60% differentiating into intermediate astrocyte precursor cells (I-APCs), 25% differentiating into a mixture of I-APCs and intermediate oligodendrocyte precursor cells (I-OPCs), and 15% differentiating into I-OPCs. The I-APCs further divided two or three times at each location to amplify the number of astrocytes [33]. La Manno et al. established the developing mouse brain atlas from samples of E7 to E18 mouse brain to visualize the spatiotemporal molecular profile. Focusing on the astrocyte clusters in the mouse brain atlas, astrocyte marker genes (*Gfap* encoding the astrocyte-specific intermediate filament protein glial fibrillary acid protein, *Agt* encoding the angiotensin receptor, and *Aqp4* encoding the aquaporin-4) were expressed around E15 concomitantly with *Egfr*, *Dll1*, *Dll3*, and *Dll4* [34]. Di Bella et al. proposed a temporal competence model describing the molecular developmental trajectories from cortical stem cells to mature neurons and glia in mouse cortex between E10.5 and P4 based on scRNA-seq data and the transcriptional similarity of pseudotime-ordered cells [35]. This model posits that neurogenic factors promote a differentiation process in which the molecular identity of pyramidal neurons becomes more similar to that of astrocytes. This notion is consistent with the observation that apical progenitor cells have a common molecular identity with pyramidal neurons and astrocytes during early development [35]. These scRNA-seq findings further suggest that there are no strictly pre-committed progenitors in the developing mouse cortex, but it is still uncertain whether fate-restricted progenitors exist among cortical progenitors [36,37].

### 2.3. Molecular Mechanisms of the Transition from Neurogenesis to Gliogenesis

The transition from neurogenesis to gliogenesis is one of the key events during brain development, and several molecular signaling pathways regulating this process have been identified, such as janus kinase signal transducer and activator of transcription (JAK-STAT), phosphatidylinositol-3 kinase (PI3K), Notch, and Smad pathways [38,39,40,41], while extrinsic signals required for induction of astrocytic genes include epidermal growth factor, bone morphogenetic proteins, leukemia inhibitory factor, and ciliary neurotrophic factor (CNTF) [42,43,44,45,46]. In cultures of single neural progenitor cells, transition was associated with activation of chicken ovalbumin upstream promoter transcription factors I and II (COUP-TFI/II) [47]. More recently, zinc finger- and BTB domain-containing protein 20 (Zbtb20) was also implicated in the regulation of the neuron–glia transition, as dysregulated expression in vivo altered the time window for astrocyte production [48].

In addition to these genetic factors and signaling pathways, epigenetic cues are also important modulators of neuron–glia transition. Hirabayashi et al. found that the transcriptional repressor polycomb group complex (PcG) binds to and epigenetically suppresses the proneural gene encoding neurogenin-1 (Ngn1), promoting a neurogenic to astrogenic fate switch in the developing cortex [49]. In addition, knockout of the histone methyltransferase enhancer of Zeste homolog 2 (Ezh2), a component of polycomb repressive complex 2 (PRC2), accelerated the onset of gliogenesis in mice [50]. High-mobility group A (HMGA) proteins also contribute to the onset of gliogenesis by inhibiting chromatin remodeling [51]. Overexpression of HMGA2 increased the expression of insulin-like growth factor 2 mRNA binding protein 2 (IMP2), and IMP2 perturbation directly affected astrocytic differentiation, suggesting IMP2 as one potential stage-dependent regulator of cortical progenitor differentiation potential [52]. The Hes family BHLH transcription factor 5 (Hes5) also regulates the neurogenesis to gliogenesis transition in cortex by suppressing expression of Hmga1/2 [53]. Other potential regulators of gliogenesis include Ngn2, Mash [54], and HMGN [55]. Collectively, these epigenetic, genetic, and molecular signaling pathways appear to regulate the number and distribution of astrocytes and neurons in cortex. Thus, dysfunction or mutation in any one of these genes or processes may lead to astrocyte-associated neurological disease.

## 3. Functions of Astrocytes in Neurovascular Coupling and the Glymphatic System

One of the essential roles of astrocytes is the maintenance of the local conditions of extracellular ions, neurotransmitters, and harmful molecules such as amyloid-β via neurovascular coupling and glymphatic system. Astrocyte dysfunction during development may induce abnormalities in this system and, thus, result in several developmental diseases.

### 3.1. Functions of Astrocytes in Neurovascular Coupling

Neurovascular coupling is an essential mechanism for maintaining local metabolic homeostasis under changing levels of neuronal activity. Briefly, neurovascular coupling increases local arteriole blood flow (and, hence, the supply of glucose and oxygen) via vasodilation to meet the greater energy requirements conferred by neuronal activity, particularly to clear excess extracellular K^+^ and synaptic glutamate accumulated from action potential and excitatory postsynaptic potential generation (Figure 1). This mechanism requires robust chemical signaling among neurons, astrocytes, and blood vessel cells, and disruption of this coupling has been linked to age-related neuropsychiatric diseases [56,57,58,59,60,61].

Neurovascular coupling relies on the precise anatomic positioning of astrocyte processes on blood vessels, mainly arteries and arterioles, and within perisynaptic spaces [56,62]. The terminal processes on vessels, termed endfeet, cover about 99% of the vessel abluminal surface [63,64], and a single astrocyte contacts nearly 100,000 synapses in rodents and up to two million synapses in humans [65,66]. Therefore, a single astrocyte can sense the local rate of synaptic transmission and, in this way, modulate synaptic transmission [67] and neurovascular coupling [56] through reciprocal communication involving extracellular ions and various neurotransmitters, including glutamate [68]. The neurons, astrocytes, and arteriole region linked via these signals collectively form a neurovascular unit, and these units may function independently to regulate local metabolic conditions.

The vasodilation of cortical penetrating arterioles is strongly reduced by inhibition of cyclooxygenase-1 (COX-1), which is expressed in perivascular astrocytes, but not by inhibition of cyclooxygenase-2 (COX-2), which is expressed in neurons [69]. Glutamate released from presynaptic boutons activates astrocytic metabotropic glutamate receptor-5 (mGluR5), which increases intracellular Ca^2+^. Elevated Ca^2+^ in turn triggers the production of epoxyeicosatrienoic acids (EETs) via cytochrome P450 2C11 epoxygenase (CYP2C11) and prostaglandin E2 (PGE2) via COX1, which both induce vasodilation [61]. Elevated Ca^2+^ levels also induce potassium release from astrocytic endfeet through large conductance calcium-dependent potassium channels (BKCa), which activate inward-rectifier potassium channels (Kir4.1) on vascular smooth muscle cells, leading to membrane hyperpolarization, muscle relaxation, and vasodilation [70]. Glutamate in the perisynaptic space is also taken by astrocytes through the glutamate/Na^+^-cotransporter to synthesize adenosine triphosphate (ATP), which is subsequently released to stimulate purinergic receptors on neurons, resulting in vasodilation of pial arterioles [71]. The astrocytic Ca^2+^ signaling also leads to the production and release of nitric oxide (NO), a powerful vasodilator of parenchymal arterioles, via nitric oxide synthase in endfeet [72]. Conversely, production of 20-hydroxyeicosatetraenoic acid (20-HETE), a metabolite from arachidonic acid (AA) released from astrocytes via cytochrome P450 4A (CYP4A), constricts vascular smooth muscle cells [61,73,74]. Additionally, large elevations in astrocytic endfeet Ca^2+^ via other pathways induce vasoconstriction [75]. Astrocytes also communicate with neurons and other astrocytes via gap junctions, which are composed of hemi-channels that allow transcellular passage of molecules less than 1.2 kDa, including ATP, inositol 1,4,5-trisphosphate (IP_3_), and Ca^2+^ [76,77]. This neurovascular coupling forms the basis for functional magnetic resonance imaging (fMRI) as described below [78,79].

Neural and vascular cells have distinct embryonic origins, but the critical importance of neurovascular coupling implies that the time courses of proliferation, migration, and terminal differentiation must be tightly regulated. Astrocytes are also required for establishing proper blood vessel density, as inhibition of astrogliogenesis leads to a significant decrease in vessel density and branching in cortex [80]. A delay in the development of neurovascular coupling may have substantial effects on postnatal brain development, as coupling appears fully complete by about 2 weeks after birth in rodents [81]. While the developmental processes for establishing neurovascular coupling are not fully understood, emerging evidence suggests that maldevelopment of the vascular unit may be associated with neuropsychiatric disease.

### 3.2. Astrocytes and the Glymphatic System

In addition to neurovascular coupling, astrocytes mediate cerebral spinal fluid (CSF) and interstitial fluid (ISF) flow through the parenchyma and Virchow–Robin space (Figure 2) as part of the glymphatic system [82] that removes soluble proteins and metabolic end-products [4,83]. The astrocyte endfeet surround the basal lamina, which extends from the Virchow–Robin space, and regulate CSF influx via aquaporin-4 (AQP4). In addition to CSF influx, the astrocytes also regulate CSF efflux by enlarging the CSF-drained perivascular space. This CSF–ISF exchange is mediated by aquaporin-4 (AQP4) channels expressed at high density on astrocyte endfeet. These water-permeable channels are also involved in rapid astrocyte volume regulation [84,85]; thus, astrocyte volume changes can be used as a marker for glymphatic function. The glymphatic system may be regulated by the autonomic nervous system, as recent studies have linked glymphatic clearance of waste molecules with vagus nerve activity [86]. The perineural spaces surrounding the cranial nerves, including the vagus, are known to provide some level of CSF drainage to peripheral lymphatics [87]. Additionally, vagal nerve stimulation enhanced the CSF penetrance of a low-molecular-weight fluorescent tracer (TxRed-3kD) [88]. Vagal nerve stimulation triggers the release of acetylcholine [89,90], noradrenaline [91,92,93], and serotonin [93], among which noradrenaline appears to be a modulator of the glymphatic system [4]. The α2-adrenoceptor agonist dexmedetomidine [94] enhanced glymphatic transport [95], while elevated brain noradrenaline resulted in shrinkage of the extracellular volume fraction and a reduction in both CSF influx and brain ISF influx [96]. Locus coeruleus-derived noradrenaline was also found to increase blood–brain barrier (BBB) permeability, leading to augmentation of ISF secretion and enhanced glymphatic function [97]. Thus, evidence strongly suggests that noradrenaline regulates glymphatic system function, although the underlying mechanisms are uncertain and the effects appear bidirectional, necessitating further study. Loss of locus coeruleus neurons is observed in Alzheimer’s disease [97], suggesting that the accumulation of pathogenic amyloid-β, a hallmark of this disease, may be exacerbated by deficient glymphatic clearance due to impaired modulation by noradrenaline.

## 4. Neuropsychiatric Disease and Astrocyte Activity

Recent developmental studies have implicated astrocyte dysfunction in Alexander’s disease, Rett syndrome, fragile X mental retardation, and epilepsy [98].

### 4.1. Alexander’s Disease

Alexander’s disease is a rare demyelinating disorder caused by mutations in the gene encoding glial fibrillary acidic protein (GFAP), the major intermediate filament protein of astrocytes [7], and it is the only known astrocyte-specific disease. Patients are usually diagnosed at around 2 years of age on the basis of developmental delay and MRI abnormalities in WM, T2 hypo-intensities, and T1 hyperintensities in the periventricular rim, and abnormal T2 signals and swelling or atrophy in the basal ganglia and thalamus [99]. A few reports have also documented neuronal loss in the CA1 pyramidal layer of the hippocampus and in the striatum, although this is not a consistent finding [99]. Transgenic mice harboring a mutant human GFAP gene exhibited hypertrophic astrocytes, astrocytic overexpression of stress-associated small heat-shock proteins, and inclusion bodies identical histologically and antigenically to the thick, elongated, worm-like bundles termed Rosenthal fibers observed in Alexander’s disease patients [100]. Knock-in mice with GFAP-R76H and -R236H mutations also developed Rosenthal fibers in the hippocampus, corpus callosum, olfactory bulbs, subpial regions, and periventricular regions [101]. In addition, these mice exhibited GFAP accumulation, which is sometimes referred to as “GFAP toxicity” [102]. Although these transgenic animals have provided some insights into the histopathological manifestations of Alexander’s disease, the pathomechanisms underlying cognitive delay are still unclear.

### 4.2. Rett Syndrome

Rett syndrome is a progressive neurodevelopmental disorder almost exclusively afflicting females caused by loss of the transcriptional repressor methyl-CpG-binding protein 2 (MeCP2) [8]. Clinical symptoms include respiratory abnormalities and cognitive impairment. Mice lacking the MeCP2 gene also demonstrated respiratory abnormalities, cognitive impairment, seizures, scoliosis, and sleeping problems [98,103,104], consistent with the human symptom profile. MeCP2 is highly expressed in neurons and may be involved in the formation of synaptic contacts and activity-dependent neuronal gene expression [105]. Astrocytes also express MeCP2, and MeCP2 deficiency in astrocytes causes significant abnormalities in the regulation of brain derived neurotrophic factor (BDNF), a ubiquitous regulator of neuronal dendritic and synaptic plasticity, and of cytokine production, suggesting that this deficit may alter brain inflammatory function [8,9,10,11]. Thus, astrocytes may drive Rett syndrome pathology via inflammatory reactions and insufficient BDNF signaling.

### 4.3. Fragile X Mental Retardation

Fragile X syndrome is caused by loss-of-function mutations in *FMR1*, the gene encoding the translational repressor fragile X mental retardation protein (FMRP) [106], resulting in inherited cognitive impairment and an autistic phenotype [107,108]. Fragile X syndrome is also characterized by a wide array of behavioral and metabolic impairments [109]. Specific deletion of *FMR1* in mouse astrocytes elevated spine density in the motor cortex and impaired motor skill learning in adulthood [12]. However, overexpression of FMRP in astrocytes was insufficient to completely rescue spinal and behavioral defects in Fmr1-KO mice, suggesting a joint astrocytic–neuronal contribution, whereby both astrocytes and neurons contribute to fragile X pathogenesis [12].

### 4.4. Epilepsy

Astrocytes regulate neuronal circuit formation, excitability, blood supply, and metabolism; therefore, disruption of any of these functions shifts the local excitatory–inhibitory balance, leading to epileptogenesis [13,14,15]. Investigations of brain specimens from patients with pharmacoresistant temporal lobe epilepsy and from epilepsy models have revealed alterations in the expression, localization, and function of astroglial K^+^ channels and water channels [110]. When gap junctions among astrocytes and neurons become uncoupled, K^+^ clearance is hindered, resulting in accumulation of extracellular K^+^ and concomitant neuronal hyperexcitability [14]. Neuronal degeneration due to injury or early-stage epilepsy can lead to the reactive transformation of astrocytes, and these reactive astrocytes have been shown to produce larger Ca^2+^ signals mediated by IP_3_R2 that contributes to epilepsy [15]. In addition, malfunction of glutamate transporters and the astrocytic glutamate-converting enzyme glutamine synthetase has been observed in epileptic tissue [110]. Impaired astrocytic glutamate clearance may arise in part due to abnormalities in metabotropic glutamate signaling [111]. For instance, astrocyte-specific conditional KO of mGluR5 slowed the rate of glutamate clearance via astrocytic glutamate transporters under high-frequency stimulation and increased the propensity for epileptogenesis [112]. Astrocyte-specific tuberous sclerosis complex 1 (Tsc1) conditional knockout mice also exhibited abnormal neuronal organization and seizures [113]. A model mouse with hippocampal astrocyte-specific Neo1 KO exhibited epileptiform spikes and elevated seizure susceptibility [114].

## 5. Microscopic Imaging of Astrocyte Development

Several neuropsychiatric diseases are associated with abnormalities in the morphology, distribution, number, and/or function of astrocytes; thus, high-resolution visualization of individual astrocytes and coupled populations is critical for elucidating the contributions of these cells to pathogenesis. Essential to reliable visualization is cell-specific labeling as astrocytes are relatively small and intermingled among larger neurons. There are two major methods for identifying astrocytes in brain: labeling of astrocyte-specific proteins and RNA by immunostaining and in situ hybridization, and transgenic expression of fluorescent proteins such as green fluorescent protein (GFP) under the control of astrocyte-specific promoters.

### 5.1. Labeling of Astrocyte-Specific Genes and Proteins

Astrocytes are frequently labeled with commercial antibodies against GFAP [115], S100b [116], aldehyde dehydrogenase 1 family member L1 (Aldh1l1) [117], Sox9 [118], Sox10, glutamate transporter 1 (GLT1), glutamate–aspartate transporter (GLAST) [119], and aquaporin-4 (AQP4) [120] among other proteins. Although immunostaining is easy and can even distinguish among astrocyte subtypes, some of these proteins are also expressed by other cell types such as oligodendrocytes (S100b) and neurons (GLT1, GLAST), whereas GFAP is undetectable in many hippocampal astrocytes [121]. Alternatively, detection of RNAs by in situ hybridization, in situ sequencing [122,123], smFISH [124,125], seqFISH [126], and osmFISH [127,128] can be used to identify astrocytes even if antibodies are unavailable for the protein product. However, RNA detection often cannot reveal cell morphology. Our knowledge of astrocyte gene expression patterns has benefited greatly in recent years from the development of RNA-seq technology. Furthermore, several online databases of astrocyte-specific gene expression are now available based on genome-wide transcriptome or proteome analyses [129].

### 5.2. Genetic Visualization Tools for Astrocytes

In 2012, Magavi et al. generated knock-in mice with a site-specific Cre recombinase linked to the *Thy-1.2* gene and demonstrated that astrocytes outnumber neurons in mouse cerebral cortex by 8.4-fold [2]. Subsequently, Cre- and tamoxifen-induced CreER lines with promoters that allow astrocyte-specific expression, such as promoters for *GFAP* [130,131], *Aldh1l1* [131,132], *Slc1a3* [131,133,134], *S100b* [135], *Slc6a11* [131], *Gjb6* [134], and *Fgfr3* [136], have been established. Other genetic tools now available include *GFAP-Flpo* [137], *GFAP-tTA* [138], and fluorescent proteins fused to astrocyte-specific proteins such as *GFAP-eGFP* [139] and *Aldh1l1-eGFP* [139,140].

### 5.3. Visualization Based on Somatic Transfection

Although stable cell and mouse lines are powerful tools for investigating the morphology, distribution, and function of astrocytes, establishing these lines is costly and time-consuming. Furthermore, these lines may differ in multiple ways from native astrocytes and wild-type mice. In contrast, infection of postnatal brain with adeno-associated viruses (AAVs) and delivery of transposon-based reporters to embryonic brain by electroporation are relatively efficient methods for labeling astrocyte-lineage cells in vivo (Figure 3) [141,142,143]. Fusing astroglial cell type-specific promoters to fluorescent proteins with piggyBac transposon can directly label targeted subpopulations permanently or within defined developmental phases [143], while recombinases driven by cell type-specific promoters can both efficiently label and alter the genome of astrocytes [144]. Hamabe-Horiike et al. revealed that the *Gfa2*-promoter labeled nearly 80% of astrocytes, the *Plp1*-promoter labeled nearly 96% of oligodendrocytes, and the *Mbp*-promoter labeled nearly 90% of oligodendrocytes at E15 electroporation, whereas the *CAG*-promoter labeled nearly 40% of neurons, 15% of astrocytes, and 8% of oligodendrocytes [143]. Thus, these results suggest that cell type-specific promoter approach is efficient to label astrocyte by in utero electroporation. Clavreul et al. used site-specific recombinase (Cre) electroporation to label astrocyte lineages in E15 embryos with multiple transposon-based fluorescent protein reporters called “MAGIC markers” [145]. Although this method labeled some neurons, astrocytes were labeled with multicolor combinations, permitting single-cell resolution [146]. This brainbow-based technique [147] combined with large-volume chromatic multiphoton serial microscopy (ChroMS) [148] successfully reveals clonal information. ChroMS relies on the integration of trichromatic two-photon excitation by wavelength mixing with automated serial block-face image acquisition, making multicolor imaging over >1 mm^3^ volumes possible [148]. The authors proposed that astrocyte clonal expansion and morphotype are determined in a nonordered manner within the local environment rather than by genetic predetermination. Another transposon-based glial-lineage tracing method called “StarTrack” has been used to reveal astroglial lineage behavior by color visualization. StarTrack, which is based on the piggyBac transposon system combined with astroglial-specific promoters such as those driving the expression of *GFAP* [149], *NG2* [150], and *Ubc* [151], can provide up to 12 combinations by color-barcoded proteins with localization tags. These direct in utero electroporation of transposon-based plasmids can label astrocytes more efficiently than postnatal brain labeling methods, although neurons can still be labeled. It is also possible to label astrocytes via postnatal electroporation around P0–P1, the time of astrocyte production, although expression efficiency is not as high [51]. We recently developed an alternative strategy for integration-coupled gene expression called “iOn switch”, which suppresses episomal expression after transfection to directly detect genomic gene expression with conventional electroporation [152]. By suppressing episomal expression, iOn gene expression is as stable and permanent as target gene expression in transgenic animals; hence, this tool holds great potential for future long-term astrocyte imaging to assess lineage fate and contributions to disease. For example, when we electroporated *^iOn^CAG∞RFP* at the E13 mouse cortex and fixed brain at P10, astrocytes were successfully labeled, whereas *CAG::GFP* labeled only DL neurons, which produced approximately E13 cortical progenitors. The other strategy established based on the piggyBac transposon combined Cre/*loxP* system, clonal labeling of neural progenies (CLoNe), labels the promoter-specific neuronal subpopulations with the multiple fluorescent protein (XFP) combinations [153].

In contrast to embryonic brain labeling by somatic transfection of plasmids, which is mainly used for lineage and clonal analyses, astrocyte labeling using AAVs in the postnatal brain is applied mainly for functional analyses. For example, Takano et al. fused AAV with Split-TurboID and combined this well-known chemo-genetic tool with the neuronal *hSyn1* promoter and astrocyte *GfaABC1D* promoter to collect proteins around synapses between neurons and astrocytes for proteome analysis. Results revealed that neuronal cell adhesion molecule (NRCAM) is highly enriched between neuron and astrocytes and involved in the formation of inhibitory synapses [154]. Furthermore, AAV can be combined with other genetic techniques, such as optogenetics and designer receptors exclusively activated by designer drugs, to investigate the functions of specific astrocytic proteins during development [155,156]. Several small molecule-based astrocyte markers, including sulforhodamine 101 (SR101) [157], b-Ala–Lys–Nε-AMCA [158], and 4-di-2-asp [159], have also been established for live cell imaging.

In summary, these visualization techniques have proven useful for functional analysis of astrocytes at both the cellular and the molecular levels, as well as for elucidating the interactions among astrocytes, neurons, and surrounding components. Methods using transgenic mice are limited to research for only mice. In contrast, somatic transfection and virus-derived visualization do not require genetically modified organisms and will, therefore, be essential for future research in other species, including primates and humans. Thus, these tools could provide fundamental insights into developmental disorders associated with astrocyte maldevelopment or dysfunction.

## 6. Potential Noninvasive and Mesoscopic Astrocyte Imaging Using MRI and Positron Emission Tomography (PET)

Most astrocytic signaling processes and functions, including the calcium oscillations, ion exchange, glutamate metabolism, and volume change, have so far been investigated using in vivo animal models or culture systems with various fluorescence labeling techniques, such as sulforhodamine101 [160] and genetically expressed Ca^2+^ indicators [161]. However, investigating astrocyte dysfunction in developmental disease patients requires “noninvasive brain imaging techniques”. MRI and PET are noninvasive techniques that have the advantages of providing insights on the function and structure of the whole brain at mesoscopic resolution (100–300 µm for rodents and 1–3 mm for humans). The challenges of noninvasive astrocyte imaging using MRI and PET have been described in previous studies. Here, we introduce several promising noninvasive approaches to measuring astrocyte activity in vivo (Figure 4).

### 6.1. Diffusion MRI

Diffusion MRI is exquisitely sensitive to changes in tissue microstructure, such as changes induced by cell swelling or shrinkage [162]. The “apparent diffusion coefficient” (ADC) was introduced along with the diffusion MRI to indicate the degree to which water diffusion is limited by structures within the brain [163] compared to normal Gaussian diffusion. This non-Gaussian diffusion is a highly sensitive indicator of pathological changes in brain microstructure resulting from tumor, stroke, or edema [164].

In addition to pathological changes, diffusion MRI appears sufficiently sensitive for physiological changes in astrocyte volume associated with neuronal activity. An early study using in vitro Aplysia Californica ganglia established a robust association among neuronal activity, subsequent cell volume change, and ADC [165,166], and several high-resolution diffusion MRI studies have supported the utility of ADC for detecting more subtle neuronal activation-induced changes in brain tissue microstructure using ex vivo hippocampal slices [167]. In the case of neuronal activity, this approach can induce transient neuronal and/or astrocyte volume increases that reduce the extracellular space (ECS) and decrease ADC (Figure 4A). The b-value is an artificial factor that reflects the strength and timing of the gradients to generate water diffusion-weighted images. Higher b-values (>1800 s/mm^2^) signify water diffusion within the smaller space involving ECS (Figure 4A).

Recently, we postulated that diffusion MRI can detect the astrocyte volume change because astrocytes undergo dramatic volume changes as a result of fluid flux associated with extracellular K^+^ release and subsequent activation of the astrocyte Na–K–Cl cotransporter (NKCC1), Kir4.1 inwardly rectifying potassium channel, and/or the Na^+^/K^+^ ATPase [168,169,170,171,172,173]. In addition, AQP4 activity may contribute to these volume changes by mediating ISF–CSF exchange [174]. Brain water mobility was decreased by AQP4 knockdown using RNA interference [175], and pharmacological blockade of AQP4 using 2-(nicotinamide)-1,3,4-thia-diazole (TGN-020) increased the ADC in several brain regions, including the hippocampus and cerebral cortex [176]. These results confirm that diffusion MRI signals can reflect changes in astrocyte volume under physiological conditions. Astrocytic volume changes are also observed in pathologic states. For instance, ischemia disturbs ionic homeostasis and induces the accumulation of neuroactive substances in the extracellular space, resulting in astrocyte swelling [177,178,179]. Astrocyte swelling is also observed during epilepsy, and the resulting reduction in extracellular space may exacerbate K^+^ and glutamate accumulation, leading to greater neuronal hyperexcitability and pathological firing [180,181]. Cortical spreading depression, a wave-like process characterized by loss of neuronal membrane potential and massive redistribution of intracellular and extracellular ions, including an increase in extracellular potassium, also reduced ADC [182]. Although the sensitivity of diffusion MRI to physiological and pathological astrocyte volume changes is well validated, the feasibility of diffusion MRI for clinical study of astrocyte dysfunction in neuropsychiatric disorders has not yet been tested.

### 6.2. Manganese-Enhanced MRI (MEMRI)

Recently, we examined the feasibility of imaging astrocytic calcium accumulation using manganese-enhanced MRI (MEMRI). Manganese is a chemical analog of Ca^2+^ and, thus, can enter neurons through Ca^2+^ channels and Na^+^/Ca^2+^ exchangers, thereby providing a measure of activity [183,184]. Astrocytes also express these calcium influx pathways, and several studies using MEMRI have found a link between Mn^2+^ accumulation and astrocytic activity mediated by glutamate synthetase, manganese superoxide dismutase, and calcium channels [185,186]. We developed a new MEMRI application for direct measurement of in vivo astrocyte–neuron interactions via hippocampal connexin 43 (Cx43) (Figure 4B) [187], a hemi-transmembrane channel that selectively passes Ca^2+^ between neurons and astrocytes. Manganese concentration in the hippocampus of Cx43 knockdown mice was enhanced compared to wild-type mice, and a pharmacological blocker of Cx43 also increased Mn^2+^ accumulation. These results indicate that in vivo Cx43-dependent functions of astrocytes under physiological and pathological conditions can be measured noninvasively using MEMRI. A significant potential limitation of MEMRI is manganese toxicity both to neurons [188,189] and to astrocytes [190]. However, MEMRI has already been investigated in humans using mangafodipir, an FDA-approved compound consisting of a Mn^2+^ chelator fused to the ligand fodipir [191,192]. This drug may provide new insights into neuron–astrocyte interactions in clinical trials. However, as image contrast depends on the release of Mn^2+^, there is still a risk of toxicity [193]; therefore, alternative manganese-based contrast agents are required for improved astrocytic labeling with greater safety.

### 6.3. Noninvasive Neuroimaging Techniques Using Neurovascular Coupling

fMRI exploits neurovascular coupling to monitor neuronal activity. When neurons are activated, the diameter of local blood vessels and the regional cerebral blood flow increase. Furthermore, the ratio of oxyhemoglobin to deoxyhemoglobin is altered due to changes in supply and consumption. An appropriate MR sequence sensitive to the magnetic susceptibility can detect these changes with relatively high temporal and spatial resolution. This is called a blood oxygenation level-dependent (BOLD) signal [194]. Functional ultrasound imaging (fUS) of the brain based on ultrafast Doppler can also measure local changes in cerebral blood volume related to neuronal activation and Ca^2+^ signaling [79,195]. In addition, resting-state fMRI can reveal the functional connectivity among anatomically separated regions by measuring the degree of synchronization of neuronal activity [196,197,198,199]. Neurovascular coupling observed by fMRI and fUS also reflects astrocytic activity as astrocytes are integral to neurovascular coupling. Furthermore, astrocytic Ca^2+^ signals are coupled to positive and negative BOLD fMRI signals in rats [200], and astrocyte activation evokes BOLD fMRI responses due to enhanced oxygen consumption [201]. Inhibition of AQP4 channels, which are abundantly expressed on perivascular astrocytic endfeet, and which function in the clearance of extracellular ions and metabolites, thus altering the BOLD response in the visual cortex to visual stimulation [176]. These imaging techniques are, therefore, potentially useful for examining the functional changes in astrocytes during development and the associations between astrocyte dysfunction and various neurological disorders.

### 6.4. PET

PET using tracers such as ^11^C-acetate, [^11^C]deuterium-l-deprenyl ([^11^C]DED), and other translocator proteins (TSPOs) is another widely employed imaging technique with potential applications for astroglial imaging in health and disease [202,203,204]. PET has an advantage for kinetic modeling and for measuring metabolism, but potentially harmful radioisotopes are required. The isotope ^11^C-acetate is metabolized in the tricarboxylic acid cycle; thus, consumption is a measure of oxidative metabolism. [^11^C]DED is an irreversible monoamine oxidase B (MAO-B) inhibitor. Astrocytes express elevated levels of MAO-B during activation; hence, this ligand can be employed as a biomarker for astrocytosis in conditions such as Alzheimer’s disease [205]. In autosomal-dominant Alzheimer’s disease carriers, astrocytosis as measured by [^11^C]DED was found to be initially high and then to decline, in contrast to the progressive increase in amyloid-β plaque load during disease progression, suggesting that astrocyte activation is restricted to the early stages of Alzheimer’s disease pathology [206]. The astroglial tracer BU99008, targeting imidazoline-2 binding sites (I2BS), has also been used to visualize reactive astrogliosis in postmortem AD brains [207]. In addition, activated astrocytes show upregulation of TSPOs, but measurement is hampered by TSPOs in cerebral blood vessels [203].

## 7. Conclusions and Perspective

In this review, we described mesoscopic and microscopic imaging techniques used to investigate the developmental lineages, functions, and structures of astrocytes. The vital functions of these cells during development and ongoing brain function suggests that dysfunction likely contributes to neurodevelopmental, neurological, and neuropsychiatric diseases. We posit that these techniques will facilitate the translation of preclinical research in animal models to human clinical research, possibly including clinical trials of interventions aimed at mitigating astrocytic dysfunction as disease treatments.

## Figures and Tables

**Figure 1 ijms-22-10312-f001:**
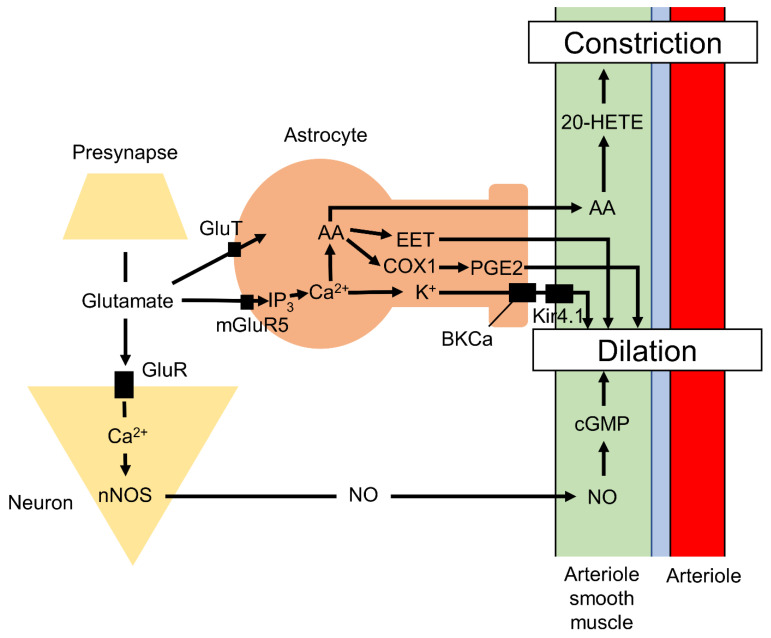
Schematic figure of the neurovascular coupling. Astrocytes and neurons intimately regulate the vasodilation and vasoconstriction of arterioles. AA, arachidonic acid; COX-1, cyclooxygenase-1; CSF, cerebrospinal fluid; EET, epoxyeicosatrienoic acids; GluT, glutamate/Na^+^-cotransporter; GluR, glutamate receptor; 20-HETE, 20-hydroxyeicosatetraenoic acid; IP_3_, inositol 1,4,5-trisphosphate; ISF, interstitial flow; Kir4.1, inward-rectifier potassium channels; NO, nitrogen oxide; nNOS, neuronal nitric oxide synthase.

**Figure 2 ijms-22-10312-f002:**
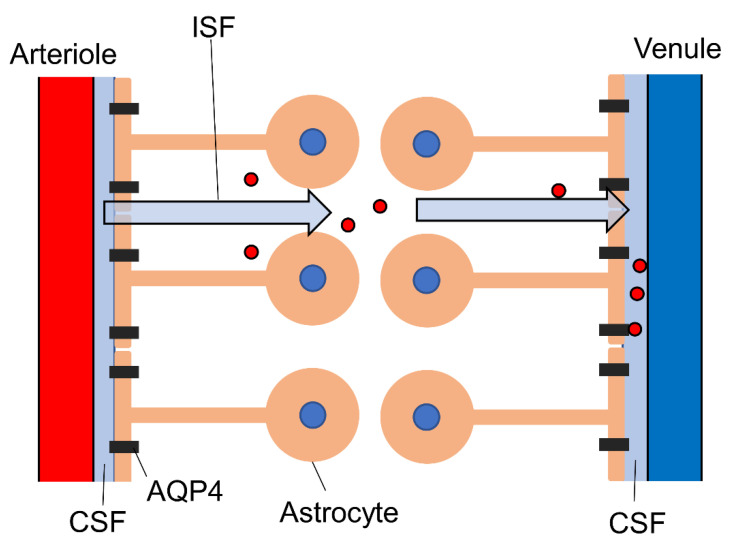
Schematic diagram of the glymphatic system. Astrocyte endfeet regulate the influx and efflux of CSF through aquaporin-4. This interstitial flow contributes to the clearance of waste, including amyloid-β and tau protein (red circles). AQP4, aquaporin-4; CSF, cerebrospinal fluid.

**Figure 3 ijms-22-10312-f003:**
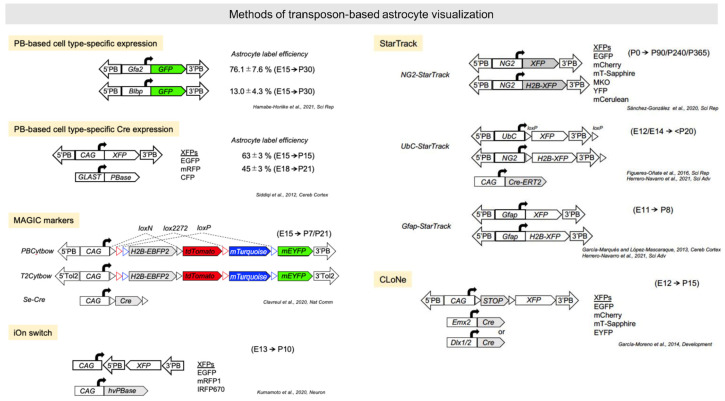
Transposon-based astrocyte visualization. Summary of genetic tools using piggyBac and Tol2 transposons. A summary of the plasmids and the timing of the experiments under investigation in the respective studies. 5′PB, 5′piggyBac terminal repeat; 3′PB, 3′piggyBac terminal repeat; 5′Tol2, 5′Tol2 terminal repeat; 3′Tol2, 3′Tol2 terminal repeat. XFPs denote each fluorescent protein included in the paper.

**Figure 4 ijms-22-10312-f004:**
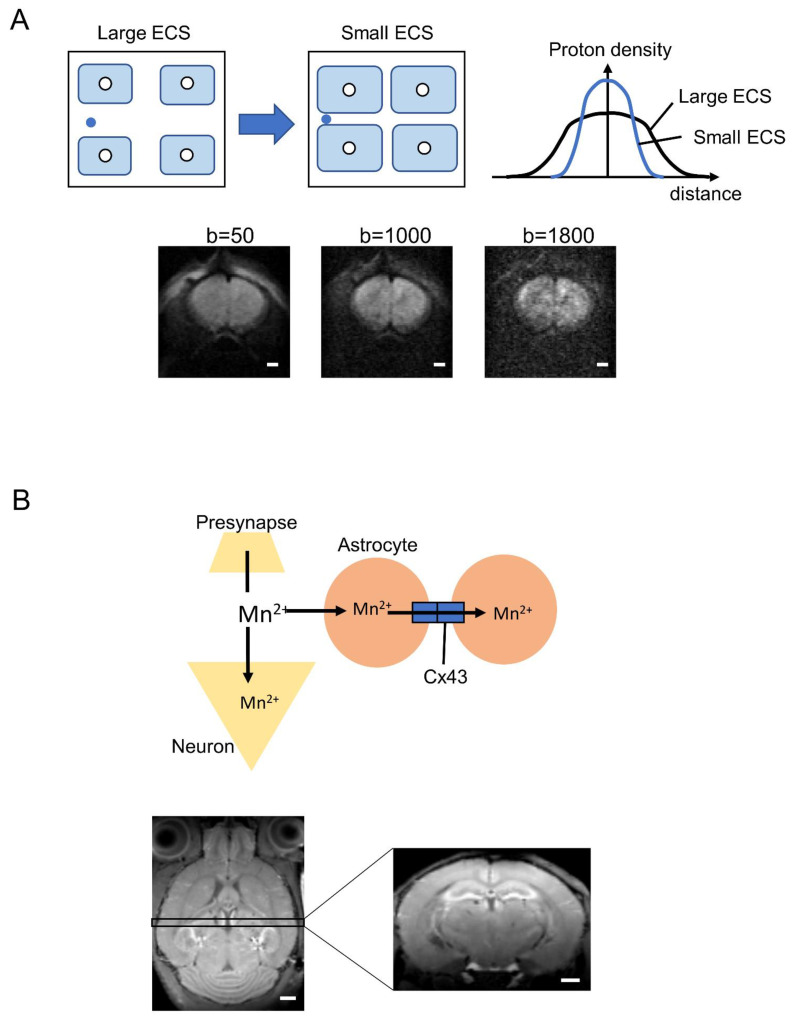
Possibility of astrocyte markers by diffusion MRI and MEMRI. (**A**) Non-Gaussian distribution of water diffusion depends on the extracellular space, which is altered by astrocyte volume change. ECS, extracellular space. Lower images are representative images of diffusion MRI with b = 50, 1000, and 1800 s/mm^2^, respectively. Images were acquired using an 11.7 T MRI system (Bruker, Germany). The b-value is an artificial factor that reflects the strength and timing of the gradients to generate water diffusion-weighted images. Higher b-values (>1800 s/mm^2^) signify water diffusion within the smaller space involving ECS. (**B**) Hypothesis of manganese accumulation in astrocytes via connexin 43 (Cx43). Lower images are representative images of MEMRI performed using an 11.7 T MRI system (Bruker, Germany). Scale bar indicates 1 mm.

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
