# Peer review of "Potential of Multiscale Astrocyte Imaging for Revealing Mechanisms Underlying Neurodevelopmental Disorders"

_ijms, 2021, doi:10.3390/ijms221910312_

Round 1

Reviewer 1 Report

The authors are thanked for this timely and comprehensive review about  the roles of astrocytes  on neuro-developmental disorders. 

I do have some minor concerns, that should they be addressed would produce a compelling paper:

  1. The authors should distinguish rodent and human astrocyte when they are discussing the development and heterogeneity of astrocytes, because  rodent astrocytes differ considerably in morphology, functionality, and gene expression. For example. in line 54-56,  Four morphologically distinct astrocytes  have been described in humans: protoplasmic astrocytes, fibrous astrocytes, interlaminar astrocytes and varicose-projection astrocytes, with the first two also present in rodents.  (PMID: 19279265).
  2.  In the section "3.2. Astrocytes and the glymphatic system", how astrocyte regulate the vascular unit and affect clearance of tau and amyloid-β was not discussed.

Reviewer 2 Report

In this manuscript, the authors describe astrocyte development and the functions of astrocytes in neurodevelopmental and neuropsychiatric diseases and also introduce some latest mesoscopic and microscopic imaging technologies for astrocyte studies. The will be very helpful. I suggest the manuscript could be accepted after some issues are addressed.

  1. The legends in Figures 1, 2, and 3 are not clear and need to be improved and clarified.
  2. In general, astrocytes have two morphological groupings: fibrous and protoplasmic (Batiuk, M.Y et al, Nat Commun 2020). The author describes 3 types in line 55. I think that is not accurate.
  3. The description for astrocyte subtypes in lines 66-69 is confusing. Please clarify it.
  4. In line 71, the author describes “there are at least three subtypes of astrocytes”, however, it is not clear about that or no ref.
  5. In line 122, “the first“ should be removed in the sentence.
  6. In ref 35, the stage of the sample is from E10.5, not E10.
  7. In line 145, if you discuss transcription factor, that means protein. “the Zbtb20 transcription factor” could be “ the transcription factor ZBTB20”. Please check the other gene/protein symbols.
  8. In line 157, “IMP2” and “Igf2bp2” mean the same gene. Please check the gene/protein symbols.
  9. In line 336/340/354, I think “S100” should be “S100b”.
  10. In line 378, please describe more details for “chromatic multiphoton serial microscopy (ChroMS)”. “ChroMS” is not shown in Figure3
  11. “CLoNe” is shown in Figure 3, however, no description in section “.5.3. Visualization based on somatic transfection”
  12. In line 412, something is missing in “-Ala-Lys-Nε-AMCA”.
  13. In line 426, remove “in” before “in vivo”.
  14. In line 445/455, “ex vivo” could be changed into “in vitro”
  15. Line 458-460, the explanation for b-value could be put in the legend.
